# In Situ Structures of the Ultra-Long Extended and Contracted Tail of Myoviridae Phage P1

**DOI:** 10.3390/v15061267

**Published:** 2023-05-29

**Authors:** Fan Yang, Liwen Wang, Junquan Zhou, Hao Xiao, Hongrong Liu

**Affiliations:** 1Institute of Interdisciplinary Studies, Key Laboratory for Matter Microstructure and Function of Hunan Province, Key Laboratory of Low-Dimensional Quantum Structures and Quantum Control, Hunan Normal University, Changsha 410082, China; yangfan@hunnu.edu.cn (F.Y.); liwenwang2020@163.com (L.W.); xiaohao201709@163.com (H.X.); 2State Key Laboratory of Infectious Disease Prevention and Control, National Institute for Viral Disease Control and Prevention, Chinese Center for Disease Control and Prevention, Beijing 100052, China

**Keywords:** bacteriophage P1, tail contraction mechanism, tail sheath protein, ultra-long tail, cryo-EM

## Abstract

The *Myoviridae* phage tail is a common component of contractile injection systems (CISs), essential for exerting contractile function and facilitating membrane penetration of the inner tail tube. The near-atomic resolution structures of the *Myoviridae* tail have been extensively studied, but the dynamic conformational changes before and after contraction and the associated molecular mechanism are still unclear. Here, we present the extended and contracted intact tail-structures of *Myoviridae* phage P1 by cryo-EM. The ultra-long tail of P1, 2450 Å in length, consists of a neck, a tail terminator, 53 repeated tail sheath rings, 53 repeated tube rings, and a baseplate. The sheath of the contracted tail shrinks by approximately 55%, resulting in the separation of the inner rigid tail tube from the sheath. The extended and contracted tails were further resolved by local reconstruction at 3.3 Å and 3.9 Å resolutions, respectively, allowing us to build the atomic models of the tail terminator protein gp24, the tube protein BplB, and the sheath protein gp22 for the extended tail, and of the sheath protein gp22 for the contracted tail. Our atomic models reveal the complex interaction network in the ultra-long *Myoviridae* tail and the novel conformational changes of the tail sheath between extended and contracted states. Our structures provide insights into the contraction and stabilization mechanisms of the *Myoviridae* tail.

## 1. Introduction

*Caudovirales*, which account for 96% of all phages, consist of an icosahedral or prolate icosahedral head, which packages the genome, and a tail [1]. The phage head is extremely stable, and its primary function is to withstand high internal pressure and protect the genome [2,3]. The phage tail is a macromolecular machine responsible for specifically recognizing host bacteria, penetrating the cell membrane, and forming a channel for genome delivery [4]. Apparently, the tail structure is a key determinant enabling phages to exert host recognition and mediate the infection process [4]. The contractile tail of *Myoviridae* phages is considered to be a complex nanodevice, and its overall structures are strongly associated with CISs [5], a class of widely distributed cell-penetrating nanomachines that mediate bacterial self-defense and exert their pathogenicity [5,6,7,8]. The conserved structural components include three main parts: a baseplate, a long inner tail tube, and a tail sheath that wraps around the tube [9]. As a model for contractile nanomachines, the *E. coli* phage T4 has been extensively studied [10,11,12,13]. During the tail-contraction, the tail sheath undergoes a massive conformational change and contracts to half and more of its original length, exposing the inner tail tube and providing the energy to drive the tail tube to eventually penetrate the target cell membrane [11,14,15]. In addition to phage T4, phage-tail-like CISs such as *P. aeruginosa* R2 pyocins [14,15], *Photorhabdus* virulence cassettes (PVC) [16], *S. entomophila* antifeeding prophage (AFP) [17], and *Algoriphagus machipongonensis* (Algo) [9] have also been presented with detailed structures, revealing many common features, but also evolving specific distinct functions. 

Phage P1, a temperate phage first discovered in 1951, transfers DNA into a wide range of gram-negative bacterial species by universal transduction and has been used to construct a fine structural genetic map of *E. coli* [18,19,20]. Interestingly, the P1 tail is exceptionally long, reaching ~2450 Å, which is 1.75 times longer than the T4 tail (~1400 Å) [21], 1.36 times longer than the jumbo myophage ΦKp24 tail (~1800 Å) [22], and 1.225 times longer than the giant myophage φKZ tail (~2000 Å) [23]. Although there have been many studies on the tail structures of *Myoviridae* phages and related CISs, current structural studies of contractile nanomachines are mostly of the single state, especially the high-resolution of the ultra-long tail, and its conformational transition in multiple states has not been fully investigated, which greatly hinders our understanding of their assembly modes and tail-sheath contraction mechanisms.

Here, we present the cryo-electron microscopy (cryo-EM) structures of the intact P1 tail in the extended and contracted states. The extended ultra-long tail of 2450 Å consists of a neck, a sixfold terminator ring, 53 sixfold tail tube rings, 53 sixfold sheath rings, and a baseplate. The length of the outer sheath of the contracted tail shrinks by about 55%, exposing the inner rigid tail tube. The local reconstructions of the extended and contracted tail were improved to 3.3 Å and 3.9 Å, respectively, allowing us to build the atomic models of the tail terminator protein gp24 [24], the tail tube protein BplB [24], and the tail sheath protein gp22 [25] in the extended state, and the atomic model of the tail sheath protein gp22 in the contracted state. The different conformations of the tube protein BplB between the top ring layer and the others, as well as the complementary rim- and plug-like structures between the adjacent tube layers, suggest the novel interaction mode of the tail tube required to maintain the stability of this ultra-long tail. Structural comparison of the extended and contracted structures reveals that the dramatic conformational changes of the sheath protein gp22 mediate the tail-contraction. Combining the new structural information with previous studies, we reveal a conserved interwoven interaction network of the sheath and the molecular mechanism of P1 tail-contraction. Our results provide rich structural information for further exploring the common features of the ultra-long contractile tail and widely distributed contractile nanomachines.

## 2. Materials and Methods

### 2.1. Sample Purification

*Escherichia coli* strain K12 (ATCC 25404) was grown in LB liquid medium (10 g tryptone, 5 g Yest extract, and 10 g sodium chloride per liter) for 4 h at 37 °C. The P1 phage (ATCC 25404-B1) was inoculated into *E. coli* K12 for 4 h at 37 °C. After the cells were lysed, the P1 phage in the supernatant was separated and collected by using low-speed centrifugation at 4000× *g* for 30 min at 4 °C. The supernatant was then precipitated with 1 M NaCl and 10% polyethylene glycol (PEG8000) (Amresco, Solon, OH, USA), and stored at 4 °C overnight. The precipitated P1 phage particles were resuspended in phage buffer (10 mM Tris-HCl, 5 mM MgCl_2_, 50 mM NaCl pH 7.4) and then were purified by a gradient density centrifugation on CsCl cushions (CsCl) (Sigma, St. Louis, MO, USA). After the centrifugation on 1.5 g/mL and 1.4 g/mL CsCl cushions at 135,000 g for 2 h at 8 °C, two phage bands were clearly visible. The lower band was collected and dialyzed against phage buffer overnight at 4 °C. The infected P1 particle band was evaluated by negative staining electron microscopy.

### 2.2. Cryo-EM Imaging

An aliquot of 3 μL from the infected P1 particle band was applied to an amorphous nickel-titanium alloy grid with carbon film, which was glow-discharged for 30 s. The grid was loaded into a Vitrobot Mark IV system (Thermo Fisher, Waltham, MA, USA) and the parameters were set as follows: temperature of 8 °C, humidity of 100%, and blotting time of 4.0 s. At the end of the sample blotting process, the grid was plunged into the liquid ethane and transferred to liquid nitrogen. Cryo-EM data were collected using a Titan Krios G3i microscope (300 keV, Thermo Fisher) equipped with a K3 summit direct electron detector (Gatan). The FEI EPU software automatically collected images of the phage P1 particles at 53,000× magnification, corresponding to a pixel size of 1.36 Å. The accumulated dose of each movie was 30 e^−^/Å^2^. A total of 7732 movies were collected; each movie stack comprised 32 image frames.

### 2.3. Image Processing and 3D Reconstruction

Intact reconstruction for the extended and contracted tails. We performed the reconstructions of the intact extended and contracted tails by using RELION software [26]. First, 17,410 intact extended tail particles were manually selected. To improve the computational efficiency, the extended tail particle images were rescaled to 512 × 512 pixels with a pixel size of 6.12 Å, and 2D classification was performed to remove the bad particles. We then performed 3D classification and refinement with C6 symmetry imposition to obtain ~30 Å resolution structure of the intact extended tail, and the reference model was generated using the “Relion_helix_toolbox” command in RELION. Using the same method as above, 6409 contracted tail particles were manually selected, and rescaled to 512 * 512 pixels with a pixel size of 2.72 Å. Using 2D classification and 3D classification and refinement, we finally obtained a 12 Å resolution structure of the intact contracted tail. The workflows of data processing are shown in Appendix A.

Local reconstruction for the extended and contracted tails. We performed the local reconstructions of the extended and contracted tail by using the RELION software [26] as follows (Appendix A): (1) A total of 40,228 tail sub-particles, including the neck and partial tail sheath region, were manually selected. (2) Two-dimensional (2D) classification was performed to obtain 32,031 extended sub-particles and 8197 contracted sub-particles. (3) Three-dimensional (3D) classification and refinement with C6 symmetry imposed was performed to obtain a 3.8 Å resolution structure of the extended tail. (4) The resolution of the extended tail structure was improved to 3.3 Å with CTF refinement. Using the same methods, the structure of the contracted tail was resolved to 3.9 Å.

### 2.4. Model Building and Refinement

Using the COOT software [27], we manually built the atomic models of proteins gp24, BplB, and gp22 for the extended tail, and protein gp22 for the contracted tail, on the basis of our cryo-EM density maps. Furthermore, we refined the models using real-space refinement, as implemented in the Phenix program [28]. For proteins with C6 symmetry, a complete model of the protein was generated using the “sym” command in UCSF Chimera [29], and the entire model was globally refined with NCS constraints [30] to separate clashing atoms between the proteins. During the iterative refinement steps, we manually checked the model to assess the quality of the refinement, and made manual adjustments until the final structure was obtained. The refinement and validation statistics are shown in Appendix A.

## 3. Results and Discussion

### 3.1. Overall Structures of the Extended and Contracted Tail of Phage P1

Phage P1 was purified from *Escherichia coli* K12 strain for cryo-EM data collection. Extended and contracted tail particles can be clearly distinguished in the cryo-EM images (Figure 1A–C), indicating that the baseplates of about 40% of the particles fell off during sample preparation (Figure 1A and Appendix A), causing the tail sheath to shrink to more than half of its original length and exposing the inner tail tube (Figure 1C). We found that all P1 particles, both extended and contracted, have a DNA-filled head, which is consistent with previous studies’ findings that P1 tail-contraction does not always result in DNA injection [18] (Figure 1A). Here, we focused on the tail only, and manually boxed 17,410 extended tail particles and 6409 contracted tail particles from a total of 7732 cryo-EM micrographs (Appendix A). Using the RELION software package [26], we reconstructed a ~30 Å resolution density map of the intact extended tail with sixfold symmetry imposed (Figure 1D), which shows that the extended P1 tail, 2450 Å long and 205 Å in diameter, is composed of three components: a neck, a trunk, and a baseplate, and that the trunk is composed of 53 rings. To improve the reconstruction resolution, we then re-boxed sub-particles and resolved the partial structure of the P1 extended tail (Appendix A), including a tail terminator ring, eight sixfold sheath rings, and eight sixfold tube rings at 3.3 Å resolution (Figure 1E and Appendix A), which allowed us to build the atomic models for the tail terminator protein gp24, the tail tube protein BplB, and the tail sheath protein gp22 (Figure 1F–H and Appendix A). The densities of the neck in our map are broken and the atomic model cannot be built. 

Using the same method as above (Appendix A), we obtain a ~12 Å resolution structure of the intact contracted tail (Figure 1I). The length of the outer sheath shrinks by about 55%, from 2100 Å to 940 Å, and the outer diameter of the helical sheath expands from 205 Å to 275 Å, allowing the inner rigid tail tube to detach from the outer sheath and be exposed to the outside (Figure 1I,J). The baseplate was not found in our contracted tail, suggesting that some degree of damage to the baseplate during sample preparation led to its detachment (Appendix A). Similarly, we resolved a partial contracted tail structure, including 15 sixfold sheath rings, at 3.9 Å (Figure 1J and Appendix A), which also allowed us to build the atomic models for the tail sheath protein gp22 (Figure 1K). The tail terminator and tail tube structures did not resolve at high resolution, probably due to the different helical parameters between the sheath and the tail tube in the contracted tail, resulting in the problem of center-alignment. Structural comparison of the sheath in the extended and contracted states indicates that the sheath protein gp22 undergoes large-scale conformational transitions (Figure 1H,K).

### 3.2. Tail Terminator Protein of Phage P1

The 53-layer tail sheath and tube of phage P1 are terminated by a hexameric terminator that covers the distal inner tail tube and embraces the top ring of the sheath (Figure 2A). The terminator consists of six copies of protein gp24 (Figure 2B,C), and each gp24 consists of four domains (Figure 2D): a central core domain (residues 23–127, 151–244), a β-hairpin domain (residues 128–150), an N-terminal domain (residues 2–22), and a C-terminal domain (residues 245–260). The six central core domains, each one consisting of seven β strands and five α helices, surround each other to form a 40 Å pore of the terminator (Figure 2C), which acts as a channel for DNA ejection [31]. The inserted β-hairpin domain extends into the distal ring of the tail tube and lies in a groove formed by two adjacent tail tube subunits, and these interactions result in the tail tube protein in the top layer having a different conformation than that in the other layers (see Figure 3G below). The N-terminal and C-terminal domains of each gp24 together form a two-antiparallel-stranded β-sheet that is inserted into the top ring of the sheath to form an augmenting β-sheet (see Figure 4F below).

Structural comparisons using the Dali server [32] revealed that the central core domains of gp24, the terminator protein gp15 of phage T4 [31], and the functional homologs proteins of *Algoriphagus machipongonensis* [9] and R2 pyocin [14] are highly conserved, but their β-hairpin domains and the extended N- and C-terminal are quite different (Figure 2E–G). We hypothesize that the structural differences between gp24 and other homologous proteins result in different interaction patterns, such as the gp15 of phage T4 interacting with the sheath protein to help maintain the integrity of the tail without the “handshake” mechanism [31]. In addition, the interaction between the β-hairpin domain of gp24 and the distal ring of the tail tube BplB is unique and may be necessary to maintain the stability of the ultra-long tail of phage P1.

### 3.3. Tail Tube Protein of Phage P1

The tail tube consists of 53 repeating 6-fold rings that form a right-handed helix with an axial rise of ~40 Å and a twist of ~20° for two adjacent rings (Figure 1E and Figure 3A). Each of the 53 rings consists of 6 copies of protein BplB and the inner and outer diameters of each ring are approximately 40 Å and 95 Å, respectively (Figure 3B). The 169-residue tail tube protein BplB consists of a β-sandwich domain flanked by an α-helix and an extended N-terminal domain (residues 1–31) arranged in an α-helix and two loops (Figure 3C). The β-sandwich is inserted by an extended β-hairpin (residues 52–77) and an O-shaped loop (O-loop, residues 113–131). This β-sandwich structure is highly conserved in the tail tube structures of *Myoviridae* and *Siphoviridae* phages and other CISs (Appendix A), such as gp19 of T4 [33], gpV_n_ of lambda [34], Pvc1 of *Photorhabdus* virulence cassettes [16] and Afp1 of *S. entomophila* AFP [17]. For the 6 BplB monomers in each tail tube ring, the six β-sandwich domains form the wall of the tube, where six α-helixes in the N-terminal domains, facing the tail terminator, form an outer rim with an inner radius of 64 Å, and the six extended β-hairpins in the β-sandwich domains, facing the tail baseplate, form a short plug with an outer radius of 58 Å (Figure 3B,D). The β-hairpin plug of one ring is precisely inserted into the N-terminal outer rim of the adjacent ring exactly, and relies on the electrostatic interactions between the plug and rim to form a stable 53-ring tail tube (Figure 3D,E). Electrostatic potential analysis indicates that the lumen of the tail tube has essentially negative electrical properties (Figure 3F) which would act as a lubricant to facilitate DNA ejection during phage infection; this electrical property is also found in the lumen of phage T4 [33]. The tail tube protein BplB interacts with the sheath protein gp22 (see Figure 4D below), which increases the stability of the ultra-long tail of phage P1.

It is noteworthy that the six copies of BplB in the top tube ring attached to the terminator exhibit striking conformational changes (Figure 3G). First, the N-terminal α-helix transforms into a β-strand that interacts with the β-hairpin domain of the terminator protein gp24 by β-sheet augmentation (Figure 3G,H). This conformational change of the N-terminal domain is also found in the *Photorhabdus* virulence cassette [16] and *S. entomophila* AFP [17] of CISs. Second, the O-loop rotates ~10° to form an additional α-helix due to lack of interactions with the tail sheath (Figure 3G).

### 3.4. Interaction Network in the Tail Sheath of Phage P1

The tail sheath consists of 53 repeated hexameric rings, forming a right-handed helical array with a length of 2100 Å, a helical rise of 40 Å and a twist of 20° (Figure 1D,E and Figure 4A,B). Unlike the sheath structures of the *Photorhabdus* virulence cassettes [16] and *S. entomophila* AFP [17] in CISs, which consist of two or more different proteins, the sheath of phage P1 consists of only one protein, gp22. According to the nomenclature of the tail sheath protein Alg2 of *Algoriphagus machipongonensis* [9], the 529-residue gp22 can be divided into a domain I (residues 426-516), a domain II (residues 35–99, and 243–425), a domain III (residues 100–242), an N-terminus (residues 3–34), and a C-terminus (residues 517-528) (Figure 4C). Structural comparison revealed that domain I and domain II of gp22 are similar to those of the sheath proteins of T4 [35], R2 pyocin [14,15], *Photorhabdus* virulence cassettes [16], *S. entomophila* AFP [17], *Algoriphagus machipongonensis* [9] and ΦKp24 [22], but the domain III is very different (Appendix A). 

Domain I of gp22, which contributes to the tail sheath wall, contains two α-helices and a β-hairpin connected by several loops. One (residues: 459–477) of the two helices interacts with the tail tube by electrostatic interactions (Figure 4B,D), which may be critical for guiding the proper assembly of the tail sheath and the tight coupling between the tail sheath and the tail tube. Domain II is the spherical projection of the sheath surface. Domain III, which extends outside the sheath, is filled with seven antiparallel β-strands connected by several loops and an α-helix. The N-terminus and C-terminus play a critical role in the sheath’s assembly and stability (Figure 4B). For the 6 gp22 monomers in each sheath ring, the N-terminus of each gp22 binds to the C-terminus of the adjacent gp22 to form a two-stranded sheet (Figure 4A,B), referred to as the “handshake” mechanism [14], and then the six handshake β-sheets in one ring attach to the 6 β-hairpins of domains I of the six gp22 proteins in the adjacent sheath ring to form 6 four-stranded β-sheets each, resulting in connections for all sheath rings (Figure 4E). Apart from the four-stranded β-sheet augmentation, there are no other obvious interactions between gp22 proteins within and between rings. These handshake and β-sheet augmentation interactions between sheath subunits are highly conserved in the contractile injection systems [9,14,15,16,17]. Notably, each of the six “handshake” β-sheets in the last sheath ring also binds to the N-terminal and C-terminal of a tail terminator protein gp24 to form a four-stranded β-sheet as well (Figure 4F). In addition, the gp22 monomers in the top sheath ring show some conformational changes with those in the other rings, including the two α-helices of the domain I, the N-terminus, and the C-terminus (Figure 4G).

### 3.5. Conformational Changes of the Contracted Tail Sheath

Tail-contraction causes the length of the tail sheath to decrease to 940 Å (Figure 1I), and the width of the sheath to increase to 275 Å (Figure 1J and Figure 5A), and the pore diameter of each sheath ring to increase from 82 Å to 108 Å, resulting in the tail tube’s detachment from the sheath (Figure 4B and Figure 5B). Vertical compression of the sheath results in a smaller helical rise of 18 Å and a larger twist of 32° (Figure 1J). The separation of the tail tube from the sheath provides the necessary conditions for the tail tube to penetrate the membrane [6]. Structural comparison of the sheath protein gp22 in the extended and contracted states revealed that the dramatic conformational changes occur at the N-terminus and C-terminus. Specifically, the C-terminus of gp22 oscillates at a small angle, whereas the N-terminus oscillates upward by nearly 100° (Figure 5C). Remarkably, the “handshake” and β-sheet augmentation mechanisms of the sheath–sheath interaction are still preserved in the contracted states (Figure 5D). Our structures confirm that the N-terminus and C-terminus of the sheath act as a hinge, while the rest of the sheath acts as a rigid-body motion [15,16,17] (Movie S1). 

The infection mechanism and the tail-contraction process of phage T4 [10,11,36] have been extensively studied and have become a paradigm of contractile injection systems. The contraction process of phage T4 propagates from the baseplate to the neck in the form of a contraction wave [11]. However, the N-terminus and C-terminus of the T4 tail sheath protein gp18, which is essential for the tail-contraction, have not been resolved in the crystal structure [35], and there are no in situ structures of the contracted sheath, which hinders our understanding of the molecular mechanisms of sheath contraction. In this paper, we have resolved the ultra-long tail structures of phage P1 in their extended and contracted states at near-atomic resolutions, allowing us to build the atomic models of the tail terminator protein gp24, the tail tube protein BplB, and the sheath protein gp22, and to illustrate the dramatic conformational changes of the sheath protein. Our in situ tail structures of P1 further confirm the molecular mechanism of tail-contraction of the *Myoviridae* phages and related CISs, and provide rich structural references for the further study of bacterial pathogenesis and drug design.

## Figures and Tables

**Figure 1 viruses-15-01267-f001:**
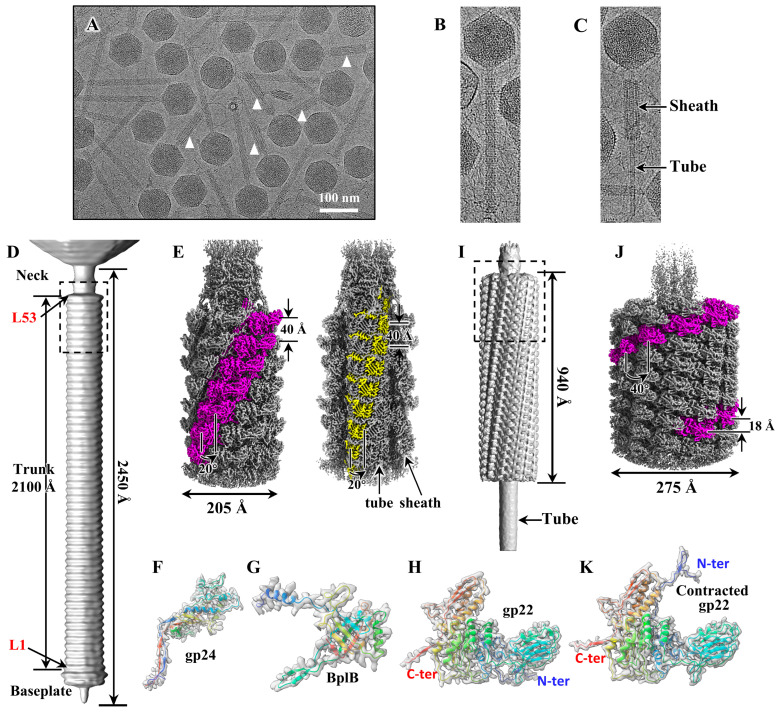
Cryo-EM image, overall structure and protein density maps of the phage P1. (**A**) Cryo-EM image of phage P1 showing the extended and contracted states. The contracted tails are marked by white triangles. (**B**) An intact extended P1 particle image showing the long and straight tail. (**C**) An intact contracted P1 particle image showing the contracted tail sheath and the exposed tail tube. (**D**) Surface view of the intact extended tail of P1. The lengths of the intact tail and tail trunk are labeled. The tail trunk consists of fifty-three rings, and the first and fifty-third rings are labeled L1 and L53, respectively. (**E**) Local reconstruction of the extended tail corresponding to the dashed box in panel (**A**) at 3.3 Å. Left: One of the six subunits of each sheath ring is presented in magenta to depict the helix assembly of the tail sheath. Right: Half of the tail sheath is removed to show the tail tube, and one of the six subunits of each tube ring is presented in yellow. (**F**–**H**) Extended density maps (gray) of the tail terminator protein gp24 (**F**), the tail tube protein BplB (**G**), and the tail sheath protein gp22 (**H**) superimposed on their atomic model (ribbon). The atomic models are shown in rainbow colors, ranging from blue at the N-terminus to red at the C-terminus, and the N-terminus and C-terminus of gp22 are labeled. (**I**) Surface view of the contracted intact tail of P1, showing that the tail sheath shrinks to 940 Å and the tail tube is exposed to the outside. (**J**) Local reconstruction of the contracted tail corresponding to the dashed box in panel (**I**) at 3.9 Å. The color codes are identical to panel (**E**) to show the helix assembly of the contracted tail sheath. (**K**) Contracted density maps (gray) of the tail sheath protein gp22 superimposed on the atomic model (ribbon). The viewing direction is identical to panel (**H**), and the N- and C-terminus of gp22 are labeled to show the conformational changes between the extended and contracted states.

**Figure 2 viruses-15-01267-f002:**
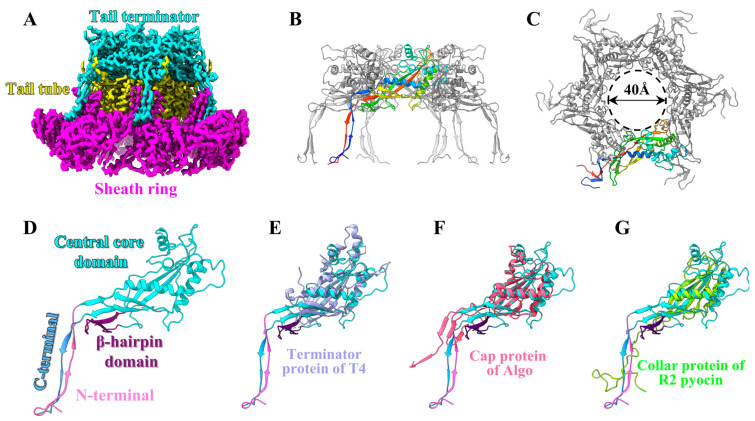
Structure of the P1 tail terminator. (**A**) The cryo-EM maps showing the interfaces between the tail terminator and the tail tube/sheath. (**B**,**C**) Side and top views of the tail terminator ring. One of the six gp24 monomers is shown in rainbow colors, ranging from blue at the N-terminus to red at the C-terminus. (**D**) Ribbon model of gp24 with the four domains labeled. (**E**–**G**) Superpositions of protein gp24 of P1 on the terminator protein of T4 (PDB ID: 3J2N), the cap protein of Algo (PDB ID:7adz) and the collar protein of R2 pyocin (PDB ID: 6u5j). The color code of gp24 is identical to that in panel (**D**).

**Figure 3 viruses-15-01267-f003:**
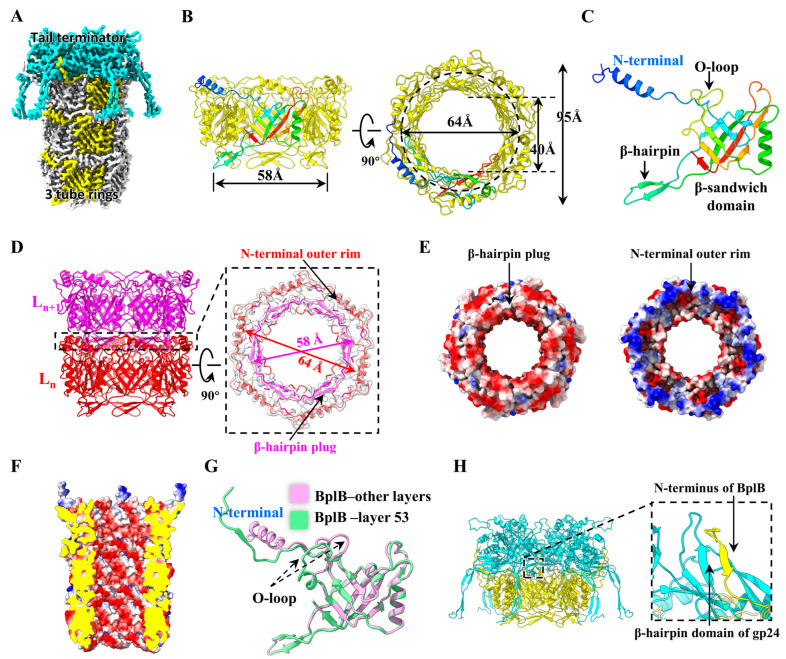
Structure of the P1 tail tube. (**A**) The cryo-EM map showing the interfaces between the tail terminator and tail tube. The tail sheath is removed to show the tail tube, and one of the six BplB monomers in each tube ring is colored yellow to show the helix assembly of the tail tube. (**B**) Side (**left**) and top (**right**) views of a tail tube ring. The inner and outer diameter of the ring are labeled. One of the six BplB monomers is shown in rainbow colors, ranging from blue at the N-terminus to red at the C-terminus. (**C**) Atomic model (ribbon) of the tail tube protein BplB. (**D**) Atomic model (ribbon) showing the interactions between two adjacent tube rings. (**E**) Electrostatic potential surfaces of the interactions between two adjacent tube rings. (**F**) The electrostatic potential surface of the inner tail tube. (**G**) Conformational changes of the tail tube protein BplB between layer 53 and other layers. (**H**) β-sheet augmentation interaction between the N-terminus of the tube protein BplB and the β-hairpin domain of the terminator protein gp24.

**Figure 4 viruses-15-01267-f004:**
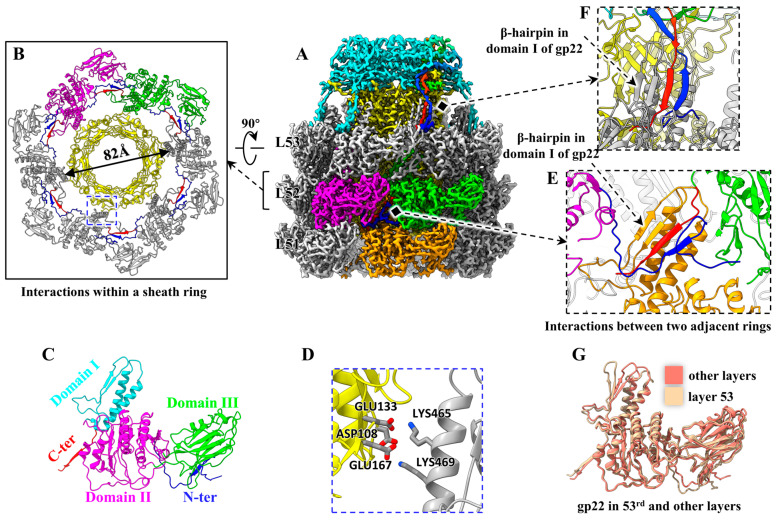
Structure of the extended tail sheath of P1. (**A**) Cryo-EM map of the tail, including the hexameric terminator (cyan), three (L51, L52, and L53) hexameric sheath rings (gray) and tube rings (yellow). The adjacent three gp22 monomers are colored in magenta, green, and orange, and the N-terminus and C-terminus of the three gp22 monomers and one gp24 monomer are colored in blue and red, respectively, to highlight the interactions. (**B**) Ribbon model showing six handshake interactions between N-terminus (blue) and C-terminus (red) in a sheath ring. The inner diameter of the sheath is labeled. (**C**) Ribbon model of the protein gp22. (**D**) Zoom-in view of the blue box in panel (**B**) to show the electrostatic interactions between the tail tube and sheath. (**E**) β-sheet augmentation interactions between two adjacent sheath rings. (**F**) β-sheet augmentation interactions between the 53rd sheath ring and the terminator. (**G**) Model superposition to show the conformational changes of the proteins gp22 in the 53rd and other layers.

**Figure 5 viruses-15-01267-f005:**
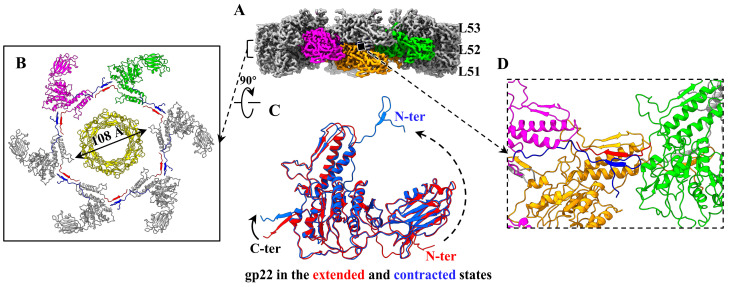
Conformational changes of the contracted P1 tail sheath. (**A**) Cryo-EM map of the three sheath rings of the contracted tail. The color codes are identical to those of Figure 4A. (**B**) Ribbon model showing six handshake interactions between N-terminus (blue) and C-terminus (red) in a contracted sheath ring. The inner diameter of the sheath is labeled. (**C**) Model superposition to show the conformational changes of the sheath proteins gp22 between extended and contracted states. (**D**) β-sheet augmentation interactions between two adjacent contracted sheath rings.

## Data Availability

The electron density maps and atomic coordinates have been deposited in the EM Data Bank (accession no. EMD-36130, and EMD-36127), and Protein Data Bank (ID codes 8JAN and 8JAJ).

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
