# Peer review of "In Situ Structures of the Ultra-Long Extended and Contracted Tail of Myoviridae Phage P1"

_viruses, 2023, doi:10.3390/v15061267_

Round 1

Reviewer 1 Report

The manuscript describes the cryoEM structure of myophage P1 tail in its extended and contractile forms at 3.3 Å and 3.9 Å, respectively. Although myophages' tails have been already decribes in the past, in particular that of phage T4, the new features reported here are the length of the tail, as well as specific details of the tail terminator (TT) and tail. This short paper is well written and up-to-date concerning the references. Comparisons with other tails TT have been performed, bt not for the MTP that should be compored to other known structures, in particular, when available the MTP conformational change at the ultimate layer.

Reviewer 2 Report

This potentially exciting paper describes a portion of phage P1 tail in the extended and contracted states. Phage P1, discovered in 1951 has one of the longest contractile tails known, so there is significant interest in deciphering the mechanisms of tail contraction. While I find the paper potentially interesting, I find it strange that the authors solely focus on a portion of the tail apparatus, missing crucial other components.

Major criticisms

1- The author state: "Phage P1 was purified from Escherichia coli K12 strain for cryo-EM data collection. Extended and contracted tail particles can be clearly distinguished in the cryo-EM images (Figure 1A-C), indicating that the baseplates of about 40% of the particles fall off during sample preparation, causing the tail sheath to shrink to more than half of its original length and exposing the inner tail tube (Figure 1C)." I'm afraid I have to disagree with this statement. The baseplate appears to move upwards with the sheath. The authors should provide evidence that the baseplate 'falls off'. I simply do not see it in Figure 1.

2 - How come the phages with a contracted tail retain DNA inside the head? This novel finding suggests the phage has some gating system preventing genome ejection upon contraction. The author should provide an explanation for this otherwise new observation.

3 - The author state: "Since the length of the intact P1 particle is more than 3000 Å , there are very few particles with intact head and tail in a cryo-EM image (Figure 1A), so we focused on the tail only, and manually boxed 17,410 extended tail particles and 6409 contracted tail particles from a total of 7732 cryo-EM micrographs (Figure S1A)". I find this statement to be false. Looking at Figure 1A, there are as many heads as tails, and the head is easier to pick than the tail. I do not understand the logic of this paper. Also, heads can be solved by applying 60-fold icosahedral symmetry, so particle number is typically not a significant problem. 

4 - Figure 5. The author talks about sheath compaction and augmentation, but the figure does not mention/show this. Can the N- and C-terminal sheath protein arms be compared? Where and how is the β-sheet augmentation achieved? 

The language is fine

Reviewer 3 Report

This paper describes the the cryo-EM structures of Myoviridae phage P1 tail in extended and contracted states. P1 presents exceptionally long tail. The authors present 3.3 Å and 3.9 Å structures of extended and contracted tails. The paper presents atomic models of the tail components. The authors show that the sheath protein undergoes substantial conformation change. The paper is well-written and referenced.

I have several concerns that I invite the authors to address:

There is no contracted state of gp24-tail terminator. Have you tried to expand symmetry of C6 aligned contracted tail particles and use non-sampling 3D class to try to parse out the particles? If you cannot align the contracted gp24, please elaborate it.

Figure 2. B-C. The rainbow coloring of the monomer will stand out if the authors use gray shade for other subunits.

Figure 3. Move the labeling (A-H) of the panels to the top to be consistent with other figures.

It is not clear that how gp22-sheath protein changes orientation after contraction. Can you analyze the motion of gp22 pre- and post-contraction in Figure 5?

Table S1. Need to provide correlation coefficient, model vs. map.

Figure S5. Need RMSD of each superimposed model and C-alphas (pruned atom pairs, can be done in ChimeraX or Chimera)

Reviewer 4 Report

The authors describe the cryoEM structures of p1 phage tails in the extended and contracted forms. By comparing the two structures, the authors reveal the role of sheath protein gp22 during the confirmation exchange. While the overall findings are interesting, there are numerous language issues.

The figure layout can be improved, the labelings are inconsistent, captions are somewhat confusing. For example, the authors the labelled the figure 3 at the bottom of each panel; E, F are hidden inside the panels in figure 4.

Discussion should be separated from the results section and the conservation of gp22 among phages should be also discussed.

In addition, the authors should consider mutating the key residues of gp22 that affect the contractibility of tail, at least in a follow up study.

 Significant language improvement is essential to improve the readability of the manuscript.  
